# DeepFRET, a software for rapid and automated single-molecule FRET data classification using deep learning

Johannes Thomsen[1], Magnus Berg Sletfjerding[1], Simon Bo Jensen[1], Stefano Stella[2], Bijoya Paul[2], Mette Galsgaard Malle[1], Guillermo Montoya[2], Troels Christian Petersen[3], Nikos S Hatzakis[1,4]*

[1]Department of Chemistry and Nanoscience Centre, University of Copenhagen, Copenhagen, Denmark; [2]Structural Molecular Biology Group, Novo Nordisk Foundation Centre for Protein Research, Faculty of Health and Medical Sciences, University of Copenhagen, Copenhagen, Denmark; [3]Niels Bohr Institute, University of Copenhagen, Copenhagen, Denmark; [4]Novo Nordisk Foundation Centre for Protein Research, Faculty of Health and Medical Sciences, University of Copenhagen, Copenhagen, Denmark

**Abstract** Single-molecule Förster Resonance energy transfer (smFRET) is an adaptable method for studying the structure and dynamics of biomolecules. The development of high throughput methodologies and the growth of commercial instrumentation have outpaced the development of rapid, standardized, and automated methodologies to objectively analyze the wealth of produced data. Here we present DeepFRET, an automated, open-source standalone solution based on deep learning, where the only crucial human intervention in transiting from raw microscope images to histograms of biomolecule behavior, is a user-adjustable quality threshold. Integrating standard features of smFRET analysis, DeepFRET consequently outputs the common kinetic information metrics. Its classification accuracy on ground truth data reached >95% outperforming human operators and commonly used threshold, only requiring ~1% of the time. Its precise and rapid operation on real data demonstrates DeepFRET's capacity to objectively quantify biomolecular dynamics and the potential to contribute to benchmarking smFRET for dynamic structural biology.

*For correspondence: hatzakis@chem.ku.dk

Competing interests: The authors declare that no competing interests exist.

## Introduction

Single-molecule Förster resonance energy transfer (smFRET) combined with TIRFm (total internal reflection fluorescence microscopy) is a key powerful method to study the structure of biomolecules and provide a dynamic perspective in structural biology (*Lerner et al., 2018*). Capturing the real-time readouts of nanometer-scale distances of individual biomolecules by smFRET allows the direct observations of dynamics, interactions, and intermediates of stochastic non-accumulating events, as well as dynamic equilibria between unsynchronized molecules, all of which are obscured in ensemble averaging techniques (*Dimura et al., 2016*; *Hellenkamp et al., 2018*; *Holmstrom et al., 2019*; *Juette et al., 2016*; *Newton et al., 2019*; *Preus et al., 2015*; *Roy et al., 2008*; *Schuler and Eaton, 2008*; *Stella et al., 2018*). The high fidelity and proficiency of smFRET established it as a key toolbox for the accurate characterization of mechanisms, biomolecular interactions function, and even structures of biomolecules (*Craggs and Kapanidis, 2012*; *Dulin et al., 2018*; *Kalinin et al., 2012*; *Kilic et al., 2018*; *Ratzke et al., 2014*), under both in vitro (*Schluesche et al., 2007*; *Sharma et al., 2008*; *Stein et al., 2011*) and in vivo (*Okamoto et al., 2020*; *Sakon and Weninger, 2010*) conditions. Despite its great quantitative utility and profound impact on structural biology, smFRET is not a direct imaging modality and data treatment for extracting quantitative dynamic information relies

**eLife digest** Proteins are folded into particular shapes in order to carry out their roles in the cell. However, their structures are not rigid: proteins bend and rotate in response to their environment. Identifying these movements is an important part of understanding how proteins work and interact with each other. Unfortunately, when researchers study the structures of proteins, they often look at the 'average' shape a protein takes, missing out on other conformations the protein might only be in temporarily.

An important technique for studying protein flexibility is known as single molecule Förster resonance energy transfer (FRET). In this technique, two light-sensitive tags are attached to the same protein molecule and give off a signal when they come into close contact. This nano-scale sensor allows structural biologists to get information from individual protein movements that can be lost when looking at the average conformations of proteins.

Advances in the instruments used to perform FRET have made observing the motion of individual proteins more widely accessible to non-specialists, but the analysis of the data that these instruments produce still requires a high level of expertise. To lower the barrier for non-specialists to use the technology, and to ensure that experiments can be reproduced on different instruments and by different researchers, Thomsen et al. have developed a new way to automate the data analysis. They used machine learning technology to recognize, filter and characterize data so as to produce reliable results, with the user only needing to perform a couple of steps.

This new analysis approach could help expand the use of single-molecule FRET to different fields , allowing researchers to investigate the importance of protein flexibility for certain diseases, or to better understand the roles that proteins have in a cell.

on multiple layers of preprocessing: raw image treatment, trace selection, and data analysis. Raw image treatment (*Greenfeld et al., 2012*; *Hon and Gonzalez, 2019*; *Juette et al., 2016*; *Preus et al., 2015*; *Stella et al., 2018*) and data analysis of the selected smFRET traces is in general well-standardized and relies on well-defined methodologies with strong theoretical backing (*Hon and Gonzalez, 2019*; *Schmid et al., 2016*).

The actual trace selection can be time-consuming but crucial due to the presence of undesired phenomena at the single-molecule scale, such as sample aggregation, fluorescent contaminants, incomplete or incorrect sample labeling, complex photophysical behaviors, and high noise, to mention a few (*Algar et al., 2019*; *Hellenkamp et al., 2018*; *Roy et al., 2008*). Existing software (*Greenfeld et al., 2012*; *Hellenkamp et al., 2018*; *Hon and Gonzalez, 2019*; *Juette et al., 2016*; *Preus et al., 2015*; *Stella et al., 2018*) by single-molecule labs can simplify the tedious and time-consuming selection of traces and were recently expanded to large datasets (*Juette et al., 2016*) albeit requiring some form of manual supervision and hyper-parameter tuning selecting the proper thresholds by an expert user. This need for human intervention could potentially be subjected to cognitive biases especially by less experienced users and could limit the expansion of smFRET to classic biology labs. The increasing expansion of smFRET to structural biology labs would benefit from rapid and benchmarked methodologies, reproducible across laboratories, with minimal human intervention. This is highlighted by several initiatives to standardize the smFRET field (*Greenfeld et al., 2012*; *Hellenkamp et al., 2018*; *Lerner et al., 2018*; *Sali et al., 2015*).

Recent advances in machine learning (ML) and specifically deep learning (DL) (*LeCun et al., 2015*), have radically improved our capacity to access and extract information from abstract and noisy inputs independently of human interventions as we (*ATLAS collaboration, 2014*) and others have shown (*Berg et al., 2019*; *Christiansen et al., 2018*; *Falk et al., 2019*; *Gómez-García et al., 2018*; *Jones, 2019*; *Ouyang et al., 2018*; *Smith et al., 2019*; *Zhang et al., 2018*). DL implementations are providing high-level robust performances and have been successfully used to analyze and augment a wide range of the fluorescence microscopy analysis pipeline including assessing microscope image quality (*Yang et al., 2018*), in-silico cell labeling (*Christiansen et al., 2018*), single-cell morphology analysis (*Berg et al., 2019*; *Falk et al., 2019*), detecting single molecules (*White et al., 2020*; *Wu and Rifkin, 2015*) and linking smFRET experiments with molecular dynamics simulations (*Matsunaga and Sugita, 2018*), amongst others (*Berg et al., 2019*; *Christiansen et al., 2018*;

*Falk et al., 2019*; *Gómez-García et al., 2018*; *Jones, 2019*; *Ouyang et al., 2018*; *Smith et al., 2019*; *Zhang et al., 2018*).

Deep learning-based analysis has several advantages over other approaches: It recognizes abstract patterns and learns useful features directly from the raw input data, which allows the implementation of analysis routines that do not require extensive data preprocessing or empirically defined rules, and thus offer reproducible and less opinionated evaluation of single-molecule data; It is significantly faster than human annotation for large single-molecule datasets; it comes close to, or outperforms human performance; and its performance is increased when increasing dataset size constituting an ideal case for evaluating the large datasets obtained from single-molecule data (*Berg et al., 2019*; *Christiansen et al., 2018*; *Falk et al., 2019*; *Gómez-García et al., 2018*; *Jones, 2019*; *Ouyang et al., 2018*; *Smith et al., 2019*; *Zhang et al., 2018*). Especially important are convolutional deep neural networks (DNN), artificial neural networks that learn to approximate the underlying function that transforms input to associated output through multiple rounds of optimization. The strength of DNN is the ability to learn arbitrarily complex functions to best recognize particular aspects of the given input data and model complex nonlinear relationships. The DNN is then able to classify data into predefined classes based on the provided training labels. While the training of a DNN is generally a computationally intensive process, once trained the final model can easily be shared and used for making predictions at almost no computational cost to end-users.

Here we provide DeepFRET, an all-inclusive analysis software with a pre-trained DNN at its core, for a rapid, objective, and accurate assessment of smFRET data for quantifying biomolecular dynamics. The fully automated analysis software operates with minimal crucial human intervention and requires only a threshold on the data quality confidence, as an initial step, to output detailed quantification of structural dynamic from raw images. This is attained by an intuitive and user-friendly interface that integrates and automates common smFRET analysis procedures (*Greenfeld et al., 2012*; *Hellenkamp et al., 2018*; *Hon and Gonzalez, 2019*; *Juette et al., 2016*; *Preus et al., 2015*; *Stella et al., 2018*) from raw image analysis and background-corrected intensity trace extraction (*Thomsen et al., 2019*), to sophisticated trace classification, statistical analysis of single-molecule data and production of publication-quality figures of dynamic structural biology insights (see Materials and methods). DeepFRET comes as a free-to-use standalone executable for both Windows and Mac users allowing end-users with limited programming skills to easily operate it. A script-based version implemented entirely in Python enables experts to adjust features pipelining the analysis specified for their needs. We anticipate DeepFRET to take full advantage of the widespread digitization and open repository of smFRET data and form a reference point setting a bar for the data quality and data classification performance metrics, offering additional benchmarking the field for dynamic structural biology.

## Results

### DeepFRET software package

DeepFRET is an open-source software package that implements a neural network model architecture for data evaluation integrating into a user-friendly platform all common procedures for smFRET analysis (*Figure 1*, *Figure 1—figure supplements 1–2*). The neural network model architecture used here is based on a deep convolutional neural network to recognize particular spatial features present in the data. The model first passes the data through several layers of convolutions of different lengths, and in the process learns to recognize which particular elements of a sample are present at different length scales, to best classify it correctly. This has previously been used to label time-series data such as electrocardiographs (*Hannun et al., 2019*) or electrical readouts in DNA sequencing (*Wick et al., 2018*). Additionally, we added a long short-term memory (LSTM) layer after the convolutional layers, as this will also help the model to learn temporality in the data and propagate to the later frames the learned information (*Karim et al., 2018*; *Oh et al., 2018*). A detailed description of the model hyperparameters and architecture can be found in the Materials and methods section.

To ensure that the predictions of DeepFRET would generalize to a wide range of experimentally observable behaviors independently of biological systems or experimental conditions, we provide a fully pre-trained DNN model. The implemented DNN is pre-trained on 150,000 simulated traces that uniformly sample all possible FRET states, their respective lifetimes, occupancies, and transition

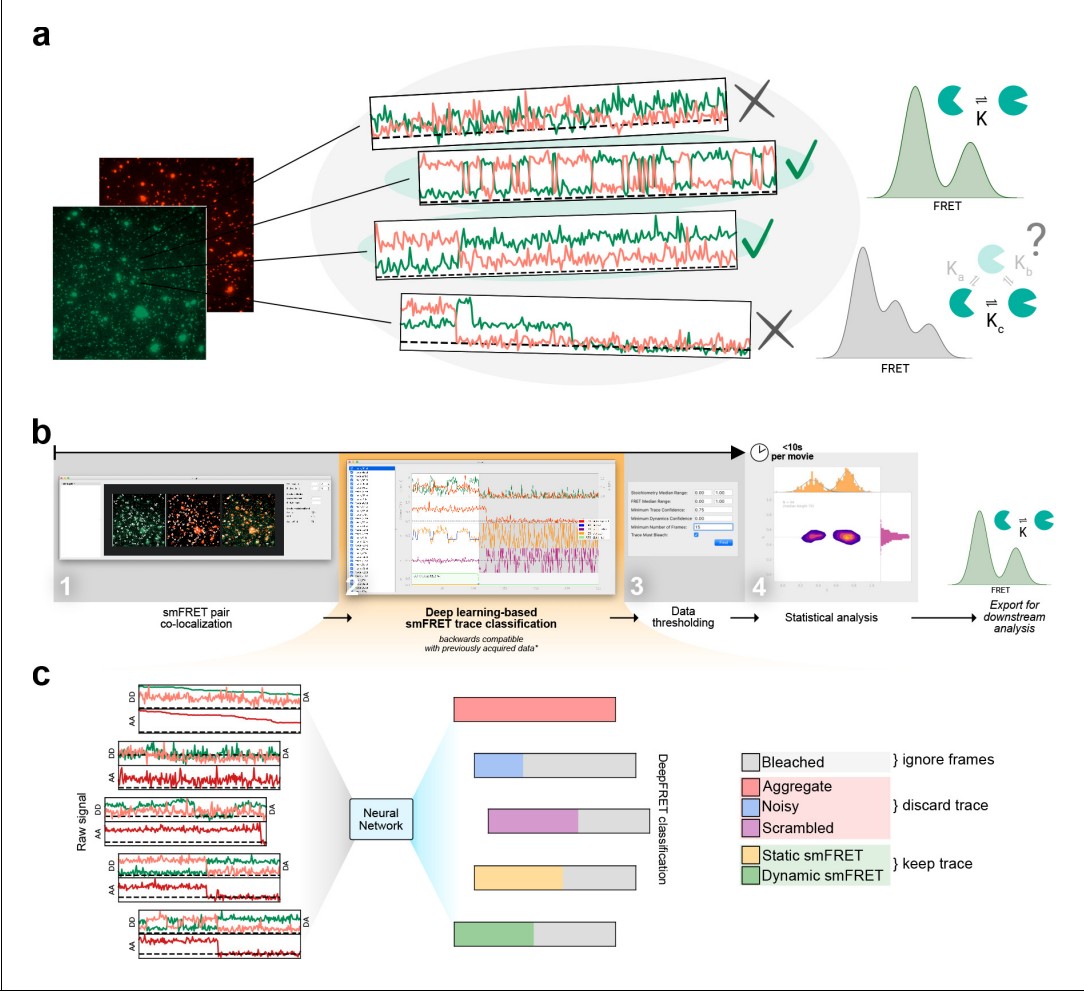

**Figure 1.** Overview of smFRET evaluation and analysis using DeepFRET. (**a**) Cartoon of the typical heterogeneous data acquired in smFRET experiments. Varying criteria for data selection for downstream analysis may yield different structural and kinetic information. (**b**) Screenshots of the provided standalone software that integrates deep learning and reduces the selection to a single-user-adjustable criterion: the DEEP FRET confidence threshold. The simple and intuitive GUI integrates all the features of our approach for rapid traces extraction from raw images to filtering of traces based on the predicted classification, treatment of smFRET data to extraction of publication-quality figures. (**c**) End-to-end sequence classification of smFRET traces by deep learning. Raw signals of donor-donor, donor-acceptor, and acceptor-acceptor intensities in the form of ASCII files can also be loaded with the DeepFRET software. The pre-trained DNN will classify individual frames into one of six different categories: bleached, static smFRET, dynamic smFRET, aggregate, noisy, and scrambled. A final smFRET confidence score is calculated, depending on each of the categories, that is used for threshold.

The online version of this article includes the following source data and figure supplement(s) for figure 1:

**Source data 1.** Data underlying *Figure 1—figure supplements 3*, *4 ,* and *7*.
**Figure supplement 1.** Schematic overview of the neural network model architecture.
**Figure supplement 2.** Data generator algorithm overview.
**Figure supplement 3.** FRET values, dynamic FRET state lifetimes, and transition pathways are uniformly sampled for all ground truth smFRET traces.
**Figure supplement 4.** Noise threshold for simulated data.
**Figure supplement 5.** Examples of randomly generated traces.
**Figure supplement 6.** Trace simulator interface.
**Figure supplement 7.** Training label dependence on the frame number.
**Figure supplement 8.** Calculation of confidence score from model predictions.
**Figure supplement 9.** Histogram interface window.
**Figure supplement 10.** Transition density window.
**Figure supplement 11.** Traces window.

pathways, as well as all possible noise levels, ensuring that the data represents all theoretically possible configurations (see *Figure 1*, *Figure 1—figure supplements 3–5*, Materials and methods for software and algorithms). As such DeepFRET does not require the selection of any direct initial guesses of FRET values or user-defined parameter pretraining. However, we do provide both a script-based method for simulating smFRET data and a simple graphical interface for expert end-users to adjust simulation distribution parameters (see *Figure 1b*, *Figure 1—figure supplement 6* and Materials and methods) for model re-training if needed (e.g. for specific circumstances or stricter criteria). This offers experts the possibility to benchmark the impact of, for example, one's sorting criteria, noise, and optical correction factors.

We built DeepFRET to treat both alternating laser excitation (ALEX) and non-ALEX FRET data. DeepFRET imports raw microscope images and performs colocalization of the two channels, to extract background corrected intensity traces of DD (donor excitation; donor emission), DA (donor excitation; acceptor emission), their respective stoichiometry, and in the case of ALEX data, also AA (acceptor excitation; acceptor emission; see *Figure 1a*, *Figure 1—figure supplement 4*). Alternatively, one can directly load and process previously-obtained time-traces ASCII format as exported from the popular software package iSMS (*Preus et al., 2015*) without their associated videos.

For a given time trace the DNN predicts and outputs six softmaxed probabilities $p_i$ to each time frame (*Figure 1c*, *Figure 1—figure supplement 5* and Materials and methods), representing the six categories it has been trained to recognize: bleached (B), static smFRET within the experimental time frame (S), dynamic smFRET (D), aggregate (A), noisy (N), and all other types of non treatable data smFRET data defined here as scrambled (X) (*Figure 1c*, *Figure 1—figure supplements 5* and *7*). Both static and dynamic traces are included for further analysis. Given these probabilities, which sums to one, a simple sliding window then searches for frames predicted by the DNN to be bleached ($p_B$ >0.5, see (*Figure 1*, *Figure 1—figure supplements 5–8*) for evaluation accuracy, and blinking exclusion). When bleaching is found, the rest of the trace is removed to exclude the photobleached frames part of a trace from further analysis. If the trace still contains a minimum number of viable frames (here set to 15 but adjustable), the probabilities are summed up over all remaining time frames for each of the five remaining categories and divided by the number of frames for normalization (see Materials and methods and *Figure 1c*, *Figure 1—figure supplements 5–8*). We define the summaries of the combined static and dynamic trace scores as the 'DeepFRET score', representing the DNN model confidence that a trace is truly smFRET. The user-friendly interface displays all the categories and their associated probabilities, and offers the option for expert users to manually revise the classified traces.

If the DeepFRET score is above the user-defined threshold, the trace is accepted for the subsequent analysis (see *Figure 1*, *Figure 1—figure supplement 8* and Materials and methods). Subsequent analysis involves two-channel fitting of idealized FRET traces using Hidden Markov modeling *HMM* (using the open-source package pomegranate); data and statistical evaluation of the abundance of FRET states and lifetimes; application of correction factors; and transition density plots (see *Figure 1b*, *Figure 1—figure supplements 6* and *9–11*). The number of underlying FRET distributions is automatically determined using Bayesian information criterion (BIC), offering the unbiased analysis of distribution of biomolecular distances (see *Figure 1b*, *Figure 1—figure supplements 6* and *9–11*). All data can be directly exported in publication-quality figures or extracted as data for user specific analysis if required.

## Performance of DeepFRET

To test our DeepFRET performance in practice, we initially compared it with commonly used threshold values. To be on the safe site, we simulated 200 ground truth smFRET traces and merged them with a dataset containing 5000 random, non-smFRET traces (too noisy, aggregates of multiple molecules, aberrant single-molecule behavior, see Materials and methods for parameter descriptions). The obtained overall FRET distribution would be akin to what one would observe experimentally before any preprocessing of smFRET data on a low purity protein sample (*Figure 2a*). Common procedures for pre-selecting valid data for treatment often rely on an initial automatic threshold for discarding this large fraction of non-smFRET data (see *Figure 1a*). This is based on any number of combinations of the anticorrelated signal of the donor and acceptor, fluorophore bleaching, noise levels, or certain ranges of fluorophore stoichiometry, if recorded using ALEX methods (*Hellenkamp et al., 2018*; *Hohlbein et al., 2014*; *Juette et al., 2016*; *Lee et al., 2005*;

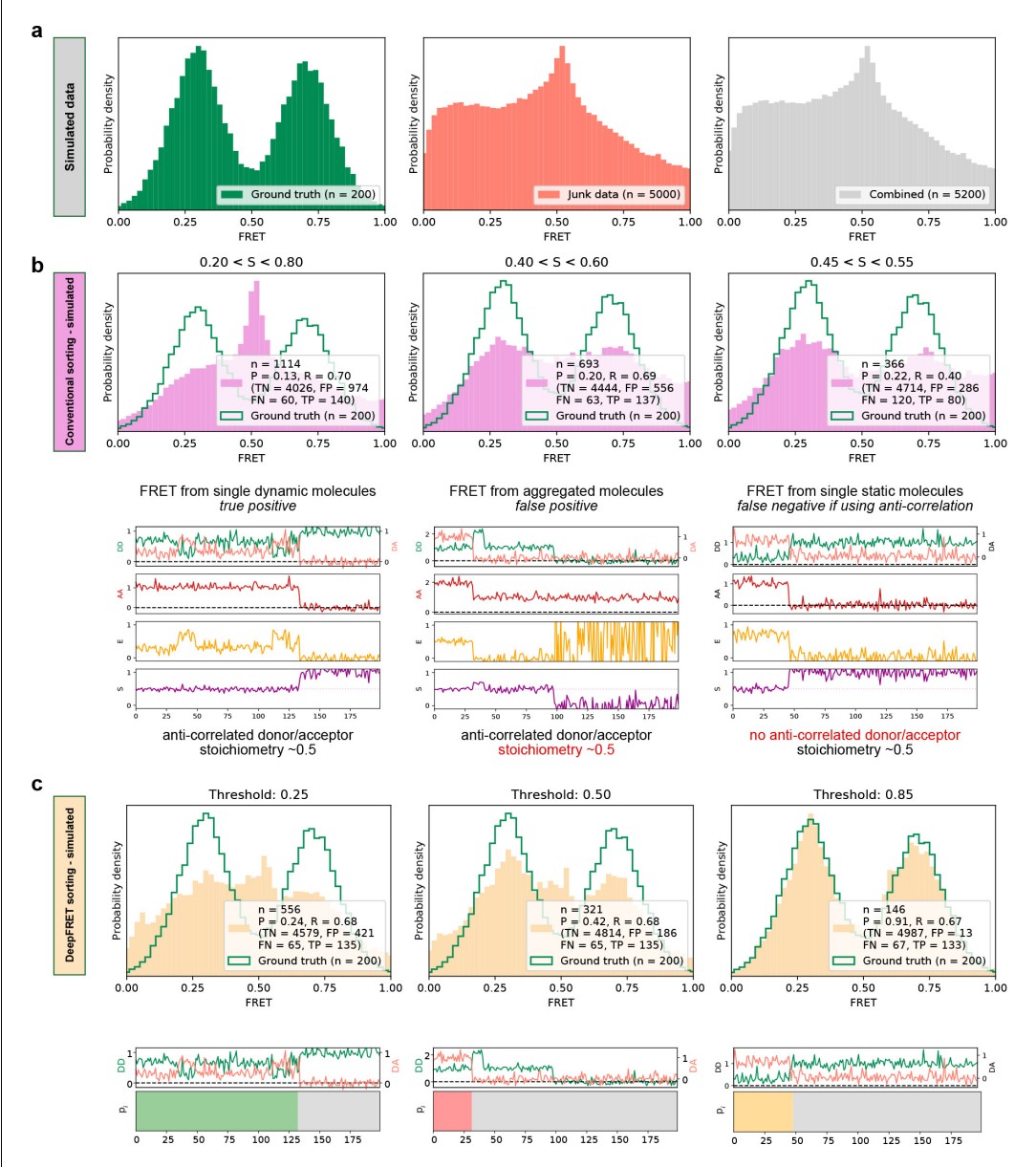

**Figure 2.** High quality FRET data evaluation. (**a**) Simulated dynamic smFRET traces transitioning between FRET states 0.3 and 0.7 (left, 'ground truth') were mixed with a larger number of traces not showing smFRET (center). The overall distribution (right, 'combined') shows how the desired data can be drowned out in non-smFRET contaminant traces. The distribution would correspond to a raw distribution as extracted from raw image analysis of smFRET on low purity protein sample before any trace selection. (**b**) Automatic selection of data based on median stoichiometry, single-molecule intensity and bleaching. The number *n* designates the number of traces accepted by the model. Tightening the selection thresholds results in slight improvement of the poor overlap of the selected data with ground truth data, highlighting the need for a time-consuming and prone to potential cognitive biases human intervention. (**c**) Automatic classification of all traces of the combined set by DeepFRET, based only on the DeepFRET score threshold variation. Even at a low threshold DeepFRET selection follows the ground truth data. Increasing the score threshold further increases the fidelity of data selection. DeepFRET correctly assigns the dynamic, bleaching and aggregate behavior on the same smFRET traces as in (b) (see *Figure 1—figure supplement 5* for more data). The single-user adjustable score threshold outperforms commonly used thresholds offering rapid, cross-lab reproducibility, and fully automatic data treatment. P: precision, R: recall, TN: true negatives, FP: false positives, FN: false negatives, TP: true positives.

The online version of this article includes the following source data and figure supplement(s) for figure 2:

**Source data 1.** Data underlying *Figure 2* and *Figure 2—figure supplements 1–3*.
**Source data 2.** Data underlying *Figure 2* and *Figure 2—figure supplements 1–3*.
**Source data 3.** Data underlying *Figure 2* and *Figure 2—figure supplements 1–3*.
*Figure 2 continued on next page*

*Figure 2 continued*

**Source data 4.** Data underlying *Figure 2* and *Figure 2—figure supplements 1–3* .
**Figure supplement 1.** Precision-recall for correctly identifying smFRET traces at different thresholds.
**Figure supplement 2.** Comparison of smFRET distribution recovery by DeepFRET at different thresholds and semi-automated methodologies under various conditions in the absence of further human intervention.
**Figure supplement 3.** Relationship between model performance and noise level.

*McKinney et al., 2006*; *Preus et al., 2015*; *van de Meent et al., 2014*). We first removed photo-bleaching and then accepted or rejected traces based on commonly used thresholds of median stoi-chiometry and max intensity (but not anticorrelation, see Materials and methods) without any manual post-inspection of the data. *Figure 2b* displays ground truth distribution (green) and the distribution of the accepted traces (pink) for varying the above thresholds. We recovered a poorly-defined FRET distribution that even at the tightest threshold does not recapitulate the underlying ground truth two-state conformational equilibrium. We calculated the common model evaluation metrics 'precision' and 'recall' (see Materials and methods) to quantify the quality of the predictions. The precision and recall, though improved by tightening the threshold, remain around 0.22 and 0.40, in the best case for simple thresholding (*Figure 2b*). The fact that out of 366 selected traces only 80 were true positive, while 286 were false positive and 120 false negative, highlights the need for human intervention as many traces are indistinguishable with simple statistical characterization (selected examples are shown below the histograms [*Figure 2b*]).

DeepFRET, on the other hand, allowed the high-fidelity recovery of the underlying ground truth distribution reaching a precision of 0.91 when setting a DeepFRET score of 0.85 (*Figure 2c*) without the need for human intervention. The virtually identical FRET distributions, matching the ground truth data, that are derived for a large fraction of score thresholds (0.5– 0.85) show no systematic biases originating from data evaluation and illustrate the minimum impact of human interventions when using DeepFRET (see *Figure 2*, *Figure 2—figure supplement 1*). As expected, the fidelity of DeepFRET pertained to correctly identifying single or complex multistate FRET distributions (see *Figure 2*, *Figure 2—figure supplements 1–2*) reaching precision 0.91 as compared to just 0.22 for standard threshold setting in the absence of further human intervention. The practically identical precision and recall for single, double or triple, state FRET distributions independently of threshold further support the wide applicability to multiple biological systems.

Quantification of precision and recall of the selection for various DeepFRET score thresholds displays the tradeoffs in recovering a high fraction of useful data. Thresholding data with scores in the regime 0.8–0.9 appears optimal for maintaining sufficient and high-fidelity data (*Figure 2*, *Figure 2—figure supplement 1*) for the trace characteristics here. Based on these data, we employed a Deep-FRET score threshold of 0.85 as optimal for maintaining high precision at reasonable recall values. We note that depending on datasets, imaging condition, noise levels, etc., users may need to adjust the threshold. The power of DeepFRET is further highlighted by the classification for the traces that were assigned as false negative and false positive by commonly accepted thresholds (traces in *Figure 2b* and *Figure 2c*, *Figure 2—figure supplement 3*). In summary, the fidelity of classification accuracy appears to supersede currently used simple thresholding, without human interventions. This was achieved in a fraction of the time required for data classification by human operators (~1 min for 10,000 traces on a normal laptop, as compared to potentially days for manually inspected traces). This improved classification was also achieved entirely without any preprocessing or post-inspection of data, illustrating the power of DeepFRET to operate without human interventions and its potential to benchmark the reproducibility of smFRET data acquisition methods for multiple bio-molecular systems across laboratories.

We quantified and displayed using confusion matrix the discordance between the ground truth data and the data selected and classified by DeepFRET (*Figure 3*). In the confusion matrices, displayed in *Figure 3*, each row represents the predicted classification of traces while each column represents the ground truth data. The high classification accuracy for the annotation of individual frames is highlighted by the clear diagonal feature. We found similar classification performance for a DNN trained on non-ALEX FRET (by a DNN with only DD and DA inputs, which we will refer to as 'ALEX-disabled'; *Figure 3* right panels) signifying the applicability of the DeepFRET approach to both ALEX and non-ALEX FRET data. The misclassification between static and dynamic smFRET

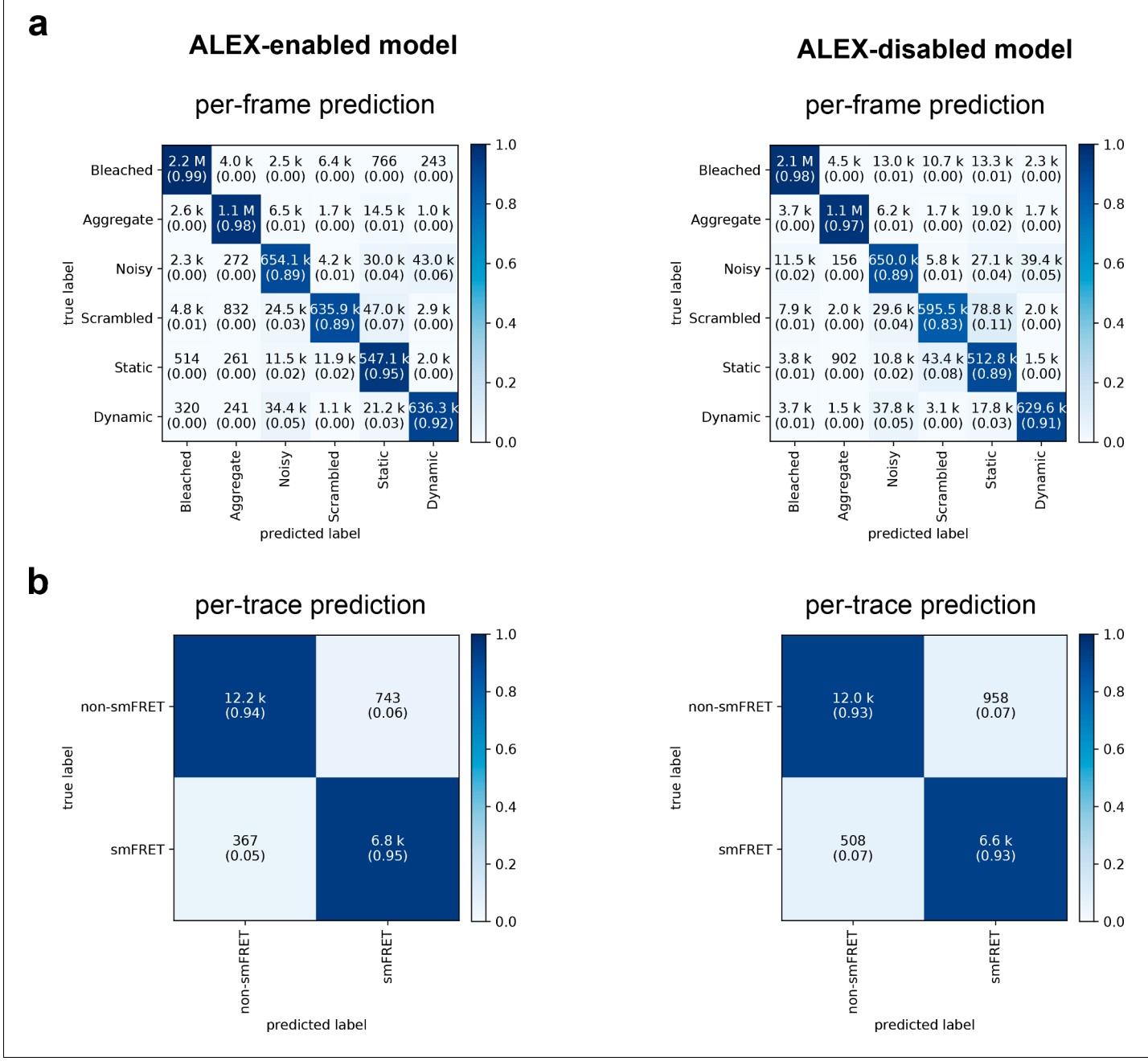

**Figure 3.** Confusion matrices of DeepFRET classification based on the ground truth data test. (**a**) Classification accuracy of data in the six categories for the ALEX-enabled model, or the ALEX-disabled model. The absolute number of frames is shown while the fractions for each classification is displayed in parentheses (as calculated row-wise for each true label). The diagonal percentages show the accurate classification of DeepFRET (**b**) per-trace classification accuracy based on accepting only traces that are classified as smFRET (static/dynamic), and non-FRET data.

The online version of this article includes the following source data and figure supplement(s) for figure 3:

**Figure supplement 1.** Precision-recall of the neural network and human participants.

**Figure supplement 1—source data 1.** Data underlying *Figure 3—figure supplement 1*.

traces is practically non-existent and consists of <3% dynamic traces being misclassified as static, for both model types. This is important for accurately quantifying the abundance of static and dynamic subpopulations within the experimental time frame, which has been shown to have a clear experimental impact (*He et al., 2019*; *Kilic et al., 2018*; *Osuka et al., 2018*; *Wood et al., 2012*).

DeepFRET was found to classify bleached or aggregated frames with a 98% the true positives for ALEX-FRET model enabled (97% for the non-ALEX model), whereas only 89% (and 83% for ALEX-disabled) of the scrambled traces were correctly classified (see also *Figure 2—figure supplement 3* for a detailed breakdown of the precision and recall). We note that the model is trained with a noise contribution that is drawn from a normal distribution of varying width (σ uniformly distributed between 0.01 and 0.30, multiplied by the maximum single fluorophore intensity) with a small contribution of gamma-distributed noise. As such, traces with σ above 0.25 are characterized by the employed DNN as 'noisy' (see *Figure 2*, *Figure 2—figure supplement 3* and Materials and methods). In a regime where transition rates are similar to the imaging temporal resolution, traces may be incorrectly classified as noisy by the model. To allow experts to accept more noisy traces or traces with fast transition rates that may appear as noisy for given imaging conditions, we integrated into the DeepFRET a visual trace simulator. This user-friendly simulator allows the generation of traces with ground truth labels of traces where all parameters are tunable to integrate the specific needs of each lab (see *Figure 1*, *Figure 1—figure supplements 6* and *11*).

We found the classification accuracy of each frame to be consistent with the classification accuracy on each trace, later derived from the overall most probable class given all predictions of individual frames of a trace (*Figure 3a*, *Figure 3b*, *Figure 1*, *Figure 1—figure supplement 6* and Materials and methods). This is achieved by adding a bidirectional long short-term memory, LSTM, layer at the end of the DNN (*Figure 1*, *Figure 1—figure supplement 1*). The LSTM layer allows coherent predictions throughout the trace and forward propagation of information detected in the first frames, such as fluorophore detection or bleaching, to the predictions for later frames. By collapsing the per-trace confusion matrix into a binary 'smFRET' and 'non-smFRET' (as shown by the cross-lines in *Figure 3b*), DeepFRET was found to be very balanced overall, with a true-positive rate of 94% for smFRET traces, and a true-negative rate of non-smFRET traces (*Figure 3c*), resulting in an overall balanced classification accuracy of 94% for the ALEX-enabled model and 93% for the ALEX-disabled model.

We then compared the classification accuracy of DeepFRET to the accuracy of three different human operators working with smFRET to evaluate the feasibility of manually inspecting and making decisions about smFRET examples. We simulated 1000 ground truth traces, of which only 46 contained actual smFRET, at different, randomly chosen levels of noise. The participants were neither informed about the underlying distribution nor the true number of smFRET traces. The test revealed that the average performance of the human operators, scoring 0.76 ± 0.10 in precision and 0.83 ± 0.14 in recall, was close to the precision-recall curve of the DNN, on a relatively small dataset (*Figure 3*, *Figure 3—figure supplement 1*). Notably, one participant scored slightly better than the model in both precision and recall but spent an average of 5 s per trace, which would significantly increase data treatment time, thus making this unfeasible in a high-throughput setting. The large

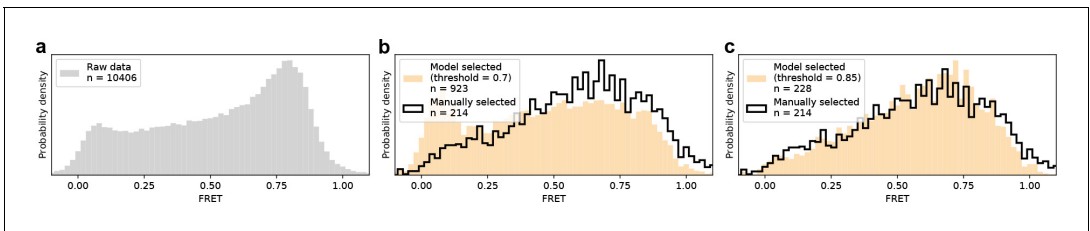

**Figure 4.** Method evaluation on real previously published smFRET data. (**a**) Raw FRET distribution as it would look before any sorting to remove incomplete or multi-labeled proteins, aggregates, cross talk, etc. (**b**, **c**) Comparison of DeepFRET data selection with published distributions for 0.7 in (**b**) and 0.85 in (**c**) thresholds. At the DeepFRET score threshold of 0.85, a high-fidelity data selection is achieved resulting in a similar distribution as compared to manual selection.

The online version of this article includes the following source data and figure supplement(s) for figure 4:

**Source data 1.** Data underlying *Figure 4* and *Figure 4—figure supplements 2–4*.
**Figure supplement 1.** DeepFRET applied to experimentally-obtained data.
**Figure supplement 2.** Distribution of trace quality of the experimental dataset.
**Figure supplement 3.** Comparison of DeepFRET, SPARTAN, iSMS, HAMMY, and ebFRET performance on simulated, ground truth data.
**Figure supplement 4.** Comparison of DeepFRET, SPARTAN, and iSMS performance on the published experimental data.

spread on precision and recall attained by human operators on these data furthermore suggest a large possible spread in experimental outcomes and highlights the advantages of unifying, reproducible methodologies independent of human interventions. We, therefore, argue that DeepFRET is equally good, or better, as careful manual inspection while offering orders of magnitude faster data evaluation.

## DeepFRET performance on real data compared to the existing robust software for smFRET analysis

The model's generalizability was initially examined by evaluation on real experimental smFRET data previously published by us (*Stella et al., 2018*). The selected published dataset contains thousands of traces that included aggregates and incomplete labeled molecules, due to low labeling efficiency. Our pre-screening (using median stoichiometry and intensity distributions) and subsequent manual inspection of the data resulted in 214 traces to exhibit smFRET. Applying our trained model with a threshold of 0.85, without any other parameter tuning, recovered 228 traces, with a FRET distribution very closely matching manual selection (*Figure 4*, *Figure 4—figure supplement 1*). The DeepFRET score of human versus machine selection displays the importance of quantitative and reproducible assessment of trace scores (*Figure 4*, *Figure 4—figure supplement 2*). The total data evaluation time of <50 ms per trace (on a recent laptop) free of human intervention highlights the potential of DeepFRET to rapidly and reliably evaluate high throughput smFRET data. Most importantly, the trace selection is deterministic, strictly relies on the score threshold, and is thus independent of potential human cognitive bias. This demonstrates the ability of the DNN to generalize to a completely new set of experimental data, without any prior expectations for signal-to-noise ratio, anti-correlation, underlying FRET distribution, etc., offering the possibility to rapidly analyze smFRET data for structural biology insights.

To further evaluate the performance of DeepFRET we compared it to the existing, robust, and widely used software packages for classification and treatment of smFRET data, iSMS, and SPARTAN (*Juette et al., 2016*; *Preus et al., 2015*), as well as ebFRET and HAMMY that focus on the kinetic analysis (*McKinney et al., 2006*; *van de Meent et al., 2014*). The performance was initially tested on simulated, ground truth data and further evaluated on published and publicly available experimental data (*Kilic et al., 2018*; *Hellenkamp et al., 2018*). For the simulated data, 200 ground truth smFRET traces were merged with 1800 simulated, non-smFRET traces and sorted by the various software packages. SPARTAN and iSMS both have sophisticated tools for automated sorting of traces (intensity, stoichiometry, and FRET thresholds in iSMS, and 26 parameter thresholds in SPARTAN including donor/acceptor correlation, SNR, intensity, FRET lifetime, and exclusion of photo-blinking), while ebFRET and HAMMY relies on simple thresholds on the intensity and FRET. Here, practically the commonly used thresholds were employed in all software packages. DeepFRET was found to sort traces at least similarly to, or better than, both SPARTAN and iSMS without any parameter tuning, closely matching the underlying ground truth distribution, while ebFRET and HAMMY would require further sorting by manual selection or additional software packages for optimal results (see *Figure 4—figure supplement 3*). We then compared the performance of the three software packages offering advanced sorting on experimental data from other groups (*Kilic et al., 2018*; *Hellenkamp et al., 2018*). To ensure proper testing, the performance was evaluated on both ALEX and non-ALEX data. Our data show that all software packages were able to reproduce published FRET distributions from raw tif files with a little discrepancy (see *Figure 4—figure supplement 4*). We used practically the default settings in both software. We note that expert users are well trained to navigate through all thresholds and define their own to further, and even more accurately, optimize data selection. This task however may become more challenging on data where the ground truth distribution is unknown and fine-tuning parameters could be subject to bias, especially for non-specialized users. The use of a single threshold and thus minimal required expertise, offered by DeepFRET may be crucial for a greater number of scientists to take advantage of this tool.

To ensure the facile operation of DeepFRET by non-machine learning experts and users without any programming skills, we provided a standalone executable along with simple and detailed instructions on how to use it (see Materials and methods). DeepFRET implements and automates in a user-friendly and intuitive platform all common procedures for smFRET analysis: sophisticated raw image analysis from raw. tiff files; particle and signal detection and localization; pixel intensity extraction for each biomolecule on both spectral channels and background corrected fluorescence and

FRET trace trajectories; automatic trace classification and sorting; unbiased analysis of number of FRET states based on BIC analysis; 2-channel fitting of idealized FRET traces using HMM analysis based on calculated number of states by BIC; data and statistical evaluation of abundance of FRET states and lifetimes; application of correction factors; and transition density plots (see *Figure 1*, *Figure 1—figure supplements 9–11*). DeepFRET furthermore offers interoperability and backward-compatible trace loading from. txt files exported from the popular iSMS software package. The software can export all results to publication-ready quality figures and also allows the extraction of data for further user-specific downstream analysis if desired.

The freely available open-source code and the underlying mathematical operations that are based on many commonly used packages (e.g. NumPy, SciPy, Matplotlib) will allow expert users to adjust features pipelining the analysis depending on their needs (see Code Availability). The DNN model is trained using Keras/TensorFlow, one of the most popular frameworks for deep learning. While the DNN is pre-trained with DeepFRET, we also provide the option for simulating new data with additional parameters offering the possibility of DNN model re-training to meet the specialized needs of trained users (e.g. multicolor FRET). The programming interface, on the other hand, allows the convenient implementation of additional scripts pipelining the analysis and to potentially expand it to additional single-molecule time-series analysis.

## Discussion

smFRET is a powerful toolkit, key for exploring dynamic structural biology, but to meet its full potential, automated standardized and user-independent analysis of data is essential. Because the experimental conditions, instruments, and biological systems drastically vary across laboratories, the treatment of data based on semi-automatic methods and simplified assumptions could yield different conclusions. DeepFRET is designed to fill this void and analyze data independently of any assumptions and reproducibly across laboratories. Our experiments show that a neural network classifier trained on purely simulated smFRET time-series accurately and efficiently recognizes and classifies smFRET both in simulated ground truth data and in a real-world dataset. DeepFRET classification accuracy consistently outperformed trace selection using commonly published thresholds. Similarly, it supersedes the selection accuracy of human operators and importantly, only requiring a fraction of the time (minutes versus weeks if traces are manually selected). Such a drastic reduction of analysis times will allow the acquisition of even larger datasets expanding the field for high throughput analysis and improving the robustness of conclusion. The fact that DeepFRET does so only requiring a score threshold, as a sole human intervention, demonstrates its strength as a novel smFRET analysis method and its potential to form a reference against which the quality of the data and the structural biology insights are benchmarked. DeepFRET was found to perform accurately for both ALEX and non-ALEX smFRET data (*Figure 4*, *Figure 4—figure supplements 3–4*) highlighting its precise classification and applicability across laboratories and methods. The limited effect of human operators on data selection on the other hand illustrates its potential to contribute to the standardization of the field, increasing reproducibility across laboratories. We anticipate that DeepFRET, combined with the advent of commercial single-molecule instruments, will contribute in materializing the smFRET as a robust mainstream toolkit for structural biology labs.

DeepFRET is currently trained and thoroughly tested to operate on 2-color ALEX and non-ALEX smFRET data. Other experimental techniques such as 3- and 4-color FRET (*Stein et al., 2011*; *Yoo et al., 2020*), protein-induced fluorescence enhancement (PIFE; *Hwang and Myong, 2014*), metal-induced energy transfer (MIET; *Chizhik et al., 2014*) or intraparticle surface energy transfer (i-SET; *Zhou et al., 2020*) could be implemented in the future through additional simulation of training data with subsequent DNN re-training and software optimization (see Materials and methods and data availability for instructions). The simulation of data in DeepFRET is based on the assumption that protein dynamics as observed by smFRET follows a Markov process. Based on this assumption, practically any kind of experimental data can be simulated, using the implemented visual trace simulator, to meet specific requirements. This may include additional noise levels above 0.30, transition probabilities larger than 0.2, traces with more than 4 FRET states, slower or faster lifetimes that may be classified as noisy, etc. In its current version, DeepFRET operates on surface-immobilized particles only, however implementing single-particle tracking as described in *Bohr et al., 2019* would allow simultaneous tracking and extraction of dynamics through smFRET on freely diffusing particles open

up exciting possibilities, for example, live cell imaging with single-particletracking (*Singh et al., 2020*) and in vivo high throughput smFRET studies (*White et al., 2020*).

DeepFRET's neural network is trained to operate for smFRET data but our approach of time-series classification and sequence annotation can conveniently be extended to consider a spectrum of stochastic single-molecule trajectories of individual turnovers including tracking (*Bohr et al., 2019*; *Ferro et al., 2019*; *Lu et al., 1998*; *Persson et al., 2013*), constant force measurements (*Goldman et al., 2015*) and blinking of individual molecules (*Durisic et al., 2014*; *Wang et al., 2019*) using either simulated or high-quality annotated experimental data for training. Consequently, we expect the neural network of DeepFRET or similar approaches to be a paradigm shift and enable new fully automated analysis methodologies related to biomolecular recognition, protein folding and dynamics, and super resolution. Such advances are paramount for increasing the breadth and impact of single-molecule studies to be fully exploited in structural biology.

## Materials and methods

We first define a nomenclature that will be used throughout the text and plots: *DD, DA, AA* (donor excitation→donor emission, donor excitation→acceptor emission, acceptor excitation→acceptor emission, respectively). A separate background signal is not considered, as we assume all model inputs to be background-corrected (i.e. background is 0).

### Synthetic smFRET data generation

Deep learning requires large amounts of diverse data in order to generalize well to unseen data. We have developed a method to generate the required thousands of ground truth traces to cover every type of empirically observed trace, with a dedicated user interface option (*Figure 1—figure supplements 2* and *6*). This algorithm includes the generation of TIRF-microscopy smFRET traces of ALEX or non-ALEX data. The traces sample any given FRET value with tunable dye photobleaching lifetime, signal noise, dye blinking, donor bleedthrough, aggregates (i.e. more than one donor/acceptor fluorophore) of any given size, as well as a 'scrambling' feature, to account for fluorophore phenomena that could not be classified as stemming from smFRET.

In order to generate traces, for each pair, we first generate the underlying FRET states from an adjustable Hidden Markov model and assume unscaled unit-intensities for *DD, DA, AA*. Then, if the energy transferred is defined by

$$FRET = DA \, / \, (DD + DA) \tag{1}$$

the remaining intensity of the donor is

$$DD = 1 - FRET \tag{2}$$

and from (1), the transferred intensity is

$$DA = -(DD^* FRET) \, / \, (FRET - 1) \tag{3}$$

In a perfectly-aligned setup, one can expect that

$$DD + DA = AA \tag{4}$$

such that the stoichiometry S will be exactly

$$S = (DD + DA) \, / \, (DD + DA + AA) = 0.5 \tag{5}$$

Initially, all fluorophores are simulated with an intensity of 1 (with absolute scaling only adjusted after applying all other parameters). Additionally, the intensity of AA should always be 1, regardless of the current FRET state. In practice, the AA intensity may not be exactly half of DD+DA (and consequently one might observe S that deviates slightly from 0.5). To account for this, we uniformly sample 'AA-mismatch' as a percentage of the unit intensity signal. Upon fluorophore photobleaching, with lifetimes sampled from an exponential distribution, either DD or DA/AA is set to 0. Noise, AA-mismatch, and donor bleedthrough are added to the ground truth signals to obtain the *observable* DD, DA, and AA, we can calculate realistic, *observable* values for E and S. For each synthetic

trace, the noise is drawn from a Normal (μ = 0, σ) distribution of varying σ. We found that, on top of the normally distributed noise, we could add the noise from a (centered) Gamma(k = 1, θ = 1.1) distribution multiplied with the noise amplitude at each frame (and is thus controlled via the noise parameter). This did not visually alter the spread of the distribution significantly but improved the robustness of predictions on real data, as we found empirically that the noise of experimental data never exactly followed a pure normal distribution.

State-of-the-art neural networks can achieve human-like or better performance on a wide range of classification tasks. Recently, however, it has been demonstrated how small modifications to the input can lead to wildly inaccurate outputs (*Goodfellow et al., 2014*). During the development of our smFRET classification model, we observed how photophysical artifacting (described as 'interesting effects' by TJ Ha's group [*Roy et al., 2008*]) would lead the model to make confident yet very inaccurate predictions to fix this, our trace generation algorithm contains extensive 'scrambling'; we found that by randomly flipping one of the channels, creating strong correlations by multiplication of the channels or adding bursts of high noise and long dark states we could avoid 'adversarial-like' predictions. We note that scrambled data is not meant to mimic observable data but instead to make the model robust against mispredictions on highly aberrant data that does not fall into the other observable categories.

We generated ground truth traces, where every frame of the sequence was labeled as one of five categories: '(B) bleached', '(A) aggregate', '(N) noisy', '(X) scrambled', '(S) static smFRET', or '(D) dynamic smFRET' (see *Figure 1—figure supplements 5–7*). Additionally, we applied label smoothing with a strength of 0.05, as this has been shown to greatly improve model robustness and prediction confidence (*Shafahi et al., 2019*).

A central element of model training is the uniform sampling of the infinite number of possible permutations of FRET data (FRET states, occupancies, lifetimes, transition pathways, and noise). For training the model, we set the following parameters (easily adjustable in the interface, see *Figure 1—figure supplement 6*):

- Up to four distinct FRET states within each trace uniformly sampled between FRET values 0 and 1 with a minimum distance of 0.1 FRET between states to be able to distinguish actual transitions from noise fluctuations. The uniformly sampled occupancies as well as FRET values are shown in *Figure 1—figure supplement 3a*.
- A transition probability from one state to another, at any frame, uniformly sampled between 0 and 0.2. In a 4-state system, the maximum sampled transition probability is thus 0.2 between each of the four states yielding a total transition probability of 0.2*(4-1)=0.6 and thus a 1–0.6 = 0.4 probability of not transitioning at any given frame. The lifetime distribution of dynamic FRET states is an evenly weighted average over the exponential decays for each possible number of FRET states and transition probabilities. A Monte Carlo simulation on 10,000 traces sampling transition probabilities uniformly between 0 and 0.2 on 2–4 state traces, verifies that our training data follows the underlying model (see *Figure 1—figure supplement 3b*).
- 0.15 probability that the trace is an aggregate.
- Transition pathway sampling of an unbiased fraction of the entire smFRET space as shown in *Figure 1—figure supplement 3c*, where a transition density plot of 1000 simulated traces is plotted displaying a random transition pathway with no measurable bias.
- 0.20 probability that a non-aggregate trace contains photoblinking.
- 0.15 probability that a trace is scrambled, and in this case 0.90 probability that the scrambling is due to incorrectly colocalized fluorophores.
- Acceptor-only mismatch between 70% and 130% of the donor intensity.
- Donor-bleedthrough between 0% and 15% the donor intensity into the acceptor channel.
- Noise drawn from a Normal(0, σ) distribution with σ values uniformly distributed between 0.01 and 0.30 (see *Figure 1—figure supplement 4*).
- 0.8 probability that the noise has an additional layer of gamma noise on top, to mimic shot noise.
- Individual trace duration of 300 frames.
- Exponentially decaying photobleaching lifetime centered around 500 frames (which will generate a fraction of traces that do not contain any photobleaching).
- 0.1 probability that the molecule will fall off the surface at a time given by an exponentially decaying lifetime centered around 500 frames (so it might not happen during the time of observation).

With these parameters, directly applicable as input for the algorithm (see Code availability), we randomly initially generated 250.000 traces of 300 frames each of randomized configurations. We then under-sampled data to balance the labels (as neural network classifiers perform worse if trained on highly class-imbalanced datasets) based on the first frame of each trace. This resulted in approximately 150,000 traces, roughly equally distributed over the five possible classes (bleaching being present in most traces naturally ends up accounting for a higher fraction of the overall frames). We used 80% for training the classifier and the remaining 20% for validation. After the training procedure, we generated an additional test set with 33.000 new traces and under-sampled it as previously, to roughly 20.000 traces, and based our statistical analysis on those alone.

We supplied only the raw features DD, DA, and AA to the model (or only DD and DA for the non-ALEX-enabled model), where for each trace, signals were normalized to the max of all signals in that trace to preserve the relative intensities between donor and acceptor. In this way, predictions done on individual smFRET traces are fully independent from every other in a given experiment, and also independent of non-standardized instrument intensity units (i.e. 'arbitrary units').

## Neural network model setup and hyperparameters

An LSTM-RNN (long short-term memory recurrent neural network) classifier was implemented in Keras with TensorFlow as backend. The structure of the network (*Figure 1—figure supplement 1*) was inspired by a recent sequence classifier for ECG time-series (*Hannun et al., 2019*) that employs both skip connections and batch normalization as means to prevent overfitting. Here we replaced the global pooling layer with stride-1 max pooling layers, and added a bidirectional LSTM layer before the final fully connected layer, which we found lead to more temporally causal and context-sensitive predictions (e.g. if the model spots multiple bleaching steps in the beginning of a trace, this information is propagated throughout, so the whole trace can be confidently marked as aggregated).

Each residual block ('Res' in *Figure 1—figure supplement 1*) contains $n = 2^x$ filters, where x is five and is incremented by 1 at every 4th block. The kernel size k starts from 16 and is reduced by 4 at every 4th Res block, so as to learn larger-scale features and gradually smaller ones. The initial convolution has the same hyper-parameters as the first residual block. A $1 \times k$ convolution is added in each residual block for efficiency (*He et al., 2016*). To avoid problems with vanishing gradients throughout such a deep model, each residual block keeps a copy of the input vector and adds it to the output vector (denoted by the '+" symbol). The long short-term memory (LSTM) cell is bidirectional and contains 16 units, and has a dropout rate of 0.4 applied to the outputs. For each frame, the outputs are distributed among six different classes by a dense layer with softmax activation.

The model loss was minimized in batches of 32 samples with the Adam optimizer, using the default parameters and the default learning rate of 0.001. The learning rate was decreased by a factor of 10 if validation loss showed no improvement over two consecutive epochs. The training was stopped early if no improvement in the validation loss was observed over five consecutive epochs. Convolutional kernels were initialized as proposed by *He et al., 2015*. Other layer configurations were left at their Keras defaults. The final model output is passed through a softmax layer, thus that for each frame the probabilities between all classes sum up to exactly 1. Further experimentation with optimizers and learning rates showed no significant improvement over the above configuration.

## Bleaching detection

In order to avoid having single-frame bleaching triggering the remainder of the trace being marked as bleached, we employ a sliding window over the whole trace. In each window, at least 4 out of 7 frames must be marked as bleached with >0.5 probability by the model. If this condition is satisfied, all frames in the window and every frame onwards is marked as bleached, and excluded from the calculation of smFRET confidence. The model predicts with >99% accuracy bleaching (*Figure 3*). Additionally, if bleaching happens faster than the first 15 frames, the whole trace is classified as bleached, regardless of model classification, as the DeepFRET score would otherwise end up being artificially inflated (see below).

For stoichiometry-based thresholding (*Figure 2*), we employed a similar sliding window but instead marked frames as bleached if the stoichiometry was outside of the range (0.3, 0.7).

## Precision and recall

We use precision and recall to quantify classifier performance. These are defined as,

$$Precision: \mathrm{P} = \mathrm{Tp} \ / \ (\mathrm{Tp} + \mathrm{Fp}) \tag{6}$$

$$Recall: \mathrm{R} = \mathrm{Tp} \ / \ (\mathrm{Fn} + \mathrm{Tp}) \tag{7}$$

where Tp, Fp, Fn are True positive, False positive, and False negative, respectively.

## DeepFRET score calculation and trace classification

In order to calculate the confidence score, probabilities for all categories for each frame are first predicted by the model, and bleached frames (see above) excluded from the score calculation. The average probability p$i$ over all frames $t$, for each of the remaining five categories is calculated, resulting in five category scores P$i$ for each category $i$.

$$P_i = \frac{\sum_t p_{i_t}}{\sum \sum_t p_{i_t}} P_i = \frac{\sum_t p_{i_t}}{\sum \sum_t p_{i_t}}$$

Static smFRET (S), dynamic smFRET (D) scores are summed into the final DeepFRET score, and aggregate (A), noisy (N), and scrambled (X) scores ignored for calculation of this (but retained and displayed for explainability for the user). See *Figure 1—figure supplements 5–8* for examples on all trace types.

## Model performance evaluation

### Noise level of synthetic data

We changed the label of traces to 'noisy' if the initial noise was drawn from a normal($\mu = 0$, $\sigma$) with $\sigma$ above 0.25. Traces above this level of noise could no longer statistically be approximated as normally distributed by D'Agostino-Pearson two-sided test for normality (*Figure 1—figure supplement 4*; which is a requirement for fitting the correct number of FRET states in a trace, using a mixture model). Although a $\sigma$ of 0.20 also fulfilled the $p<0.05$ test statistic, we chose to opt for a limit of 0.25, as we found that the neural network would otherwise tend to discard less noisy data too frequently.

### Trends in human versus machine selection

To test for differences in the way a human versus our trained model would select traces, three participants partook in the manual selection of generated data (*Figure 3—figure supplement 1*), similar to that of *Figure 2*, only this time with 1000 traces, wherein 46 were true smFRET traces and 954 non-usable traces. The number of true smFRET traces and underlying distributions were unknown to the participants.

### Performance test and comparison with existing software

For testing simple thresholding versus DeepFRET (*Figure 2*, *Figure 2—figure supplements 1–2* and *Figure 4—figure supplement 4*) we generated data with the following parameters:

- Acceptor-only mismatch between 70% and 130% of the donor intensity.
- Donor-bleedthrough of 5% of the donor intensity into the acceptor channel.
- Noise drawn from a normal($\sigma = 0.11$) distribution
- 1 (0.5 FRET), 2 (0.3, 0.7 FRET), or 3 FRET states (0.2, 0.5, 0.8 FRET)
- Transition probability of 0.1 between states, at each frame.

Other parameters were set to the same value as what is used to generate training data. Furthermore, all generated ground truth traces that did not bleach were discarded.

Our definition of 'simple thresholding' is based on single-molecule intensity, median stoichiometry, and the presence of bleaching. Here we chose not to use any values for anti-correlation as this assumes that all molecules of interest are equally dynamic, when smFRET studies have shown that this may not always be the case (*He et al., 2019*; *Kilic et al., 2018*; *Osuka et al., 2018*).

## Extra features of the software platform

### Hidden Markov model and statistical analysis

The DeepFRET GUI has the option to fit traces with a Hidden Markov model, with adjustable strictness on the number of states, according to recent best practices for smFRET data analysis, including the ability to switch between predicting states from raw fluorescence intensities or EFRET values (*Kelly et al., 2012*). We fit the traces using the pomegranate implementation of the Baum-Welch algorithm (*Schreiber, 2018*). We further provide the option to predict state values directly from the Markov Model or from the median of the classified frames for each trace, to maintain compatibility and comparability with current results in the field. We provide clustering of subsequent transition density plots, lifetime estimates with detection of degenerate states, and publication-ready plots for Pearson's correlation coefficients, DD/DA histograms, and EFRET-stoichiometry histograms.

The Hidden Markov model was verified on externally available data from the kinSoft challenge, as well as simulated data produced within DeepFRET.

## Data availability

All data used for model training and instructions on how to use it, is available at https://github.com/hatzakislab/DeepFRET-Model (*Thomsen, 2020*; copy archived at https://github.com/elifesciences-publications/DeepFRET-Model).

## Code availability

We provide DeepFRET as an accessible GUI for everyone, as well as the Python source code for expert users. The code for the GUI as well as compiled executables, with instructions for how to edit and recompile the GUI is located at https://github.com/hatzakislab/DeepFRET-GUI.

## Acknowledgements

We thank Soeren SR Bohr and Henrik Pinholt for fruitful discussions. GM and NSH are members of the Integrative Structural Biology Cluster (ISBUC) at the University of Copenhagen.

## Additional information

### Funding

| Funder | Grant reference number | Author |
| --- | --- | --- |
| Carlsbergfondet | CF16-0797 | Johannes Thomsen<br>Simon Bo Jensen<br>Nikos S Hatzakis |
| Velux Fonden | 10099 | Nikos S Hatzakis |
| Velux Fonden | 18333 | Magnus Berg Sletfjerding<br>Mette Galsgaard Malle<br>Nikos S Hatzakis |
| Novo Nordisk | NNF14CC0001 | Stefano Stella<br>Bijoya Paul<br>Guillermo Montoya<br>Nikos S Hatzakis |
| Novo Nordisk | NNF0024386 | Guillermo Montoya |
| Novo Nordisk | NNF17SA0030214 | Guillermo Montoya |
| Novo Nordisk | NNF18OC0055061 | Guillermo Montoya |

The funders had no role in study design, data collection and interpretation, or the decision to submit the work for publication.

### Author contributions

Johannes Thomsen, Conceptualization, Data curation, Software, Formal analysis, Validation, Investigation, Visualization, Methodology, Writing - original draft, Writing - review and editing; Magnus

Berg Sletfjerding, Data curation, Software, Formal analysis, Validation, Methodology, Writing - review and editing; Simon Bo Jensen, Data curation, Formal analysis, Validation, Investigation, Visualization, Writing - review and editing; Stefano Stella, Bijoya Paul, Data curation, Investigation, Writing - review and editing; Mette Galsgaard Malle, Validation, Writing - review and editing; Guillermo Montoya, Resources, Data curation, Supervision, Funding acquisition; Troels Christian Petersen, Resources, Data curation, Software, Formal analysis, Supervision, Funding acquisition, Validation, Investigation, Visualization, Methodology, Writing - original draft, Writing - review and editing; Nikos S Hatzakis, Conceptualization, Resources, Data curation, Supervision, Funding acquisition, Validation, Investigation, Visualization, Methodology, Writing - original draft, Project administration, Writing - review and editing

## Author ORCIDs
Johannes Thomsen (iD) https://orcid.org/0000-0002-0148-9114
Magnus Berg Sletfjerding (iD) http://orcid.org/0000-0001-8669-4039
Simon Bo Jensen (iD) https://orcid.org/0000-0002-8946-3831
Stefano Stella (iD) http://orcid.org/0000-0002-9078-4659
Mette Galsgaard Malle (iD) http://orcid.org/0000-0003-3722-502X
Nikos S Hatzakis (iD) https://orcid.org/0000-0003-4202-0328

## Decision letter and Author response
Decision letter https://doi.org/10.7554/eLife.60404.sa1
Author response https://doi.org/10.7554/eLife.60404.sa2

## Additional files
### Supplementary files
• Transparent reporting form

### Data availability
All data sets used for figures are provided as source data in the manuscript Source code and executable can be found at https://github.com/hatzakislab/DeepFRET-Model/ (copy archived athttps://github.com/elifesciences-publications/DeepFRET-Model), https://github.com/hatzakislab/DeepFRET-GUI. All source data are also available at: https://sid.erda.dk/sharelink/AOEC0wxxXO.

The following previously published datasets were used:

| Author(s) | Year | Dataset title | Dataset URL | Database and Identifier |
|---|---|---|---|---|
| Kilic S, Felekyan S, Doroshenko O, Boichenko I, Dimura M, Vardanyan H, Bryan LC, Arya G, Seidel CAM, Fierz B | 2018 | Single-molecule FRET reveals multiscale chromatin dynamics modulated by HP1$\alpha$ | https://doi.org/10.5281/zenodo.1069675 | Zenodo, 10.5281/zenodo.1069675 |
| Hellenkamp B, Schmid S, Doroshenko O, Opanasyuk O, Kühnemuth R, Rezaei Adariani S, Ambrose B, Aznauryan M, Barth A, Birkedal V, Bowen ME, Chen H, Cordes T, Eilert T, Fijen C, Gebhardt C, Götz M, Gouridis G, Gratton E, Ha T, Hao P, Hanke CA, Hartmann A, | 2018 | Precision and accuracy of single-molecule FRET measurements-a multi-laboratory benchmark study | https://doi.org/10.5281/zenodo.1249497 | Zenodo, 10.5281/zenodo.1249497 |

Hendrix J, Hildeb-
randt LL, Hirschfeld
V, Hohlbein J, Hua
B, Hübner CG,
Kallis E, Kapanidis
AN, Kim JY, Krai-
ner G, Lamb DC,
Lee NK, Lemke EA,
Levesque B, Levitus
M, McCann JJ,
Naredi-Rainer N,
Nettels D, Ngo T,
Qiu R, Robb NC,
Röcker C, Sanabria
H, Schlierf M,
Schröder T, Schuler
B, Seidel H, Streit
L, Thurn J, Tinne-
feld P, Tyagi S,
Vandenberk N,
Vera AM, Weninger
KR, Wünsch B,
Yanez-Orozco IS,
Michaelis J, Seidel
CAM, Craggs TD,
Hugel T

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
