## [Decision Letter]

**Acceptance summary:**

DeepFRET, a new deep learning-based platform for standardized, automated, and unbiased single-molecule FRET data analysis, has great potential to lower the threshold for smFRET expertise, allowing for a greater number of scientists to take advantage of this powerful technique.

**Decision letter after peer review:**

Thank you for submitting your article "DeepFRET: Rapid and automated single molecule FRET data classification using deep learning" for consideration by *eLife*. Your article has been reviewed by two peer reviewers, and the evaluation has been overseen by Sebastian Deindl as the Reviewing Editor and Suzanne Pfeffer as the Senior Editor. The following individual involved in review of your submission has agreed to reveal their identity: Shixin Liu (Reviewer #1).

The reviewers have discussed the reviews with one another and the Reviewing Editor has drafted this decision to help you prepare a revised submission.

Summary:

A major bottleneck in many single-molecule FRET experiments is the need to sort through and classify the single-molecule time traces in order to separate the good traces in an experiment from ones that contain artifacts. In this work, Thomsen et al. describe a new platform for standardized, automated, and unbiased single-molecule FRET data analysis. The software is based on deep learning and is intended to act as the sole tool required to go from smFRET data acquisition all the way to quantitative analyses and publication-quality figure production. Rapid and automated analysis is enabled following a single user-input parameter (quality threshold), ensuring minimal user intervention in the data analysis process. A user-friendly GUI is provided to further simplify analysis. Importantly, the platform is open source written in Python, so users may adjust the code for their specific needs.

If successful, this platform could lower the threshold for smFRET expertise, allowing for a greater number of scientists to take advantage of this powerful tool. However, in its current form the manuscript could benefit from important clarifications to convince the readers of the novelty and superiority of this platform compared to the numerous previously published smFRET data analysis software packages. One weakness of the validation is that it was done only on a single real data set from one experiment performed by their lab.

Essential revisions:

1) It is unclear how pre-training the model can account for the infinite possible FRET states/lifetimes/occupancies/transition pathways/noise etc. How can this not bias the results to look for traces that are similar SNR etc. to the training data? For example, the 0.2 max probability of transition between states in the training dataset could bias analysis toward long-lived FRET states. The authors should comment on this.

2) Comparison of DeepFRET to human accuracy in picking "clean traces" does not seem to be an appropriate comparison (and is obviously faster). Manual trace selection is generally no longer a standard means to analyze smFRET data given the freely available open-source automated alternatives (e.g. HAMMY, ebFRET, SPARTAN, etc.). The comparison to other available software packages is important to convince users of the superior or at least equivalent performance of DeepFRET in automated trace selection. The authors should include such a comparison in the revised version of the manuscript.

3) Along the same lines, a better way to prove DeepFRET's trace analysis power is to take several datasets and compare analyses from HAMMY, ebFRET etc. vs. DeepFRET. The authors should include such comparative analyses on more than one dataset in the revised version of the manuscript.

4) It would be helpful if, in the Discussion section, the authors could provide a discussion of the tool's limitations. A discussion that talks about cases where their tool might fail would be useful for researchers who want to use their tool or build upon it. For example, in the discussion, the Materials and methods section notes that photophysical effects that are sometimes observed in smFRET experiments can be problematic for the method (it seems like the tool would likely classify these as non-useful traces even though they might reflect the "true" signal from the experiment [e.g. observation of PIFE in work from TJ Ha's group]).

---

## [Author Response]

Essential revisions:1) It is unclear how pre-training the model can account for the infinite possible FRET states/lifetimes/occupancies/transition pathways/noise etc. How can this not bias the results to look for traces that are similar SNR etc. to the training data? For example, the 0.2 max probability of transition between states in the training dataset could bias analysis toward long-lived FRET states. The authors should comment on this.

This is indeed a very valid comment and has been central for the process as there is an infinite number of possible permutations of FRET data, and improper training of the model could lead to biased trace selection. We thank the reviewers for allowing us to further comment on this as it may not have been clear in the manuscript.

To introduce as little bias as possible towards specific FRET states, we strived to generate a representative fraction of the entire smFRET space sampling uniformly a large number of the infinite possible permutations of data (*FRET states/lifetimes/occupancies/transition pathways/noise)*. To verify this and further characterize the training data, we took a series of cautious steps which are outlined below:

a) *FRET states and occupancies*. The number of FRET states in a given trace was sampled uniformly from 1 to 4 states. Each state was randomly assigned a FRET value by uniform sampling between 0 and 1. For traces with more than one state, a minimum distance of 0.1 FRET was required between states (i.e. random, uniform sampling was repeated until the requirement was fulfilled) to be able to distinguish actual transitions from noise fluctuations. We verified that all FRET states as well as occupancies of the 150 k traces in the training data were uniformly distributed between 0 and 1 as displayed in Figure 1—figure supplement 3A. We realised that this may not have been clear in the original submission, so we have detailed this further in the Materials and methods subsection “Synthetic smFRET data generation”, in the revised version. Also, we corrected the legend of Figure 1—figure supplement 3A to “smFRET traces were generated with 1-4 randomly defined FRET states […]” as due to a phrasing error it stated that traces were generated with either 1, 2 or 3 states in the original submission.

b) *Transition probability and lifetimes.* We thank the reviewers for noticing the phrasing error in the original submission “with below 0 and 0.2 probability of transitioning”. The transition probability for each frame of the trace, from a given state to another, is uniformly sampled *between* 0 and 0.2. The transition probabilities are relevant *between* states, such that in a 4-state system with maximum allowed transition probabilities of 0.2, the combined transition probability at any given frame is 0.2*(4-1 states)=0.6. On the other hand, in a 2-state system with 0.01 transition probabilities between states, the combined transition probability at any given frame is 0.01*(2-1 states)=0.01. The mean lifetimes of the above mentioned systems are thus 1/0.6=1.7 frames and 1/0.01=100 frames, respectively. The lifetime distribution of the entire training data is an evenly weighted average over the exponential decays for each possible number of FRET states and transition probabilities as shown in Figure 1—figure supplement 3B. A Monte Carlo simulation on 10,000 traces sampling transition probabilities uniformly between 0 and 0.2 on 2-4 state traces, verifies that our training data follows the underlying model (Figure 1—figure supplement 3B). Hence, we sample a broad range of transition probabilities uniformly covering both long- and short-lived FRET states, striving to resemble most experimental data without introducing bias towards specific lifetimes. Acknowledging that this may not have been clear in the manuscript, we have added the new Figure 1—figure supplement 3B with lifetime distributions and the Monte Carlo simulation together with a brief description of how these were derived in the Materials and methods subsection “Synthetic smFRET data generation”.

c) *The transition pathway* follows a Markov chain which is randomly generated from a transition matrix with probabilities sampled as described above. The Markov chain of each simulated trace is generated using the Hidden Markov Model implementation of the opensource Python package named pomegranate. To verify that the training data sample a subset of all possible transition pathways uniformly, we have plotted a transition density plot of n=10,000 simulated traces. This convincingly illustrates a completely random and uniform transition pathway as expected. Acknowledging that this critical information was not implemented in the original submission, we have added the plot as a new Figure 1—figure supplement 3C, and made further clarifications in the Materials and methods subsection “Synthetic smFRET data generation”.

d) *Noise level.* Experimental data can sample a wide range of noise levels dependent on the biological system, instrumentation, experimental setup etc. To cover a broad range of possible noise levels, we took the following steps: First, donor and acceptor intensities of each trace were generated from the underlying true FRET values. Then, noise was added to the intensities by sampling from a normal distribution with σ values uniformly distributed between 0.01 and 0.30. To mimic shot noise, an additional layer of gamma noise was added on top with 0.8 probability. To justify the chosen range of σ values, we had plotted simulated FRET distributions at various noise levels as shown in Figure 1—figure supplement 4. The broad range of noise levels that we sample in the training data resemble most experimental data without introducing bias towards specific SNR due to uniform sampling supporting our training data and thus the model is unbiased towards specific SNR within the given range of σ values. We have clarified in the revised version both in the subsection “Performance of DeepFRET” and Discussion that in a regime where transition rates are similar to the imaging temporal resolution, dynamic smFRET traces may be incorrectly classified as noisy by the model. Trace simulation and model re-training (or changing imaging settings) would solve this.

We acknowledge that despite our rigorous attempts to introduce as little bias as possible by sampling all parameters uniformly, specialised users may have a better judgement and knowledge of their specific systems, noise levels, state lifetimes or transition probabilities amongst other parameters. When training a model, there might be some kind of bias defining the interval limits of the sampled parameters. If therefore, an expert user wants to tailor the model to better meet their specific needs, DeepFRET implements a user-friendly trace simulation interface where new FRET traces can easily be simulated based on user-defined parameters (see Figure 1—figure supplement 6) and used for retraining of the DNN model following our instructions (https://github.com/hatzakislab/DeepFRETModel). We have further highlighted this in the new paragraph in the Discussion section.

2) Comparison of DeepFRET to human accuracy in picking "clean traces" does not seem to be an appropriate comparison (and is obviously faster). Manual trace selection is generally no longer a standard means to analyze smFRET data given the freely available open-source automated alternatives (e.g. HAMMY, ebFRET, SPARTAN, etc.). The comparison to other available software packages is important to convince users of the superior or at least equivalent performance of DeepFRET in automated trace selection. The authors should include such a comparison in the revised version of the manuscript.

We wish to highlight that the main the scope of this manuscript is to provide an intuitive platform requiring minimal human intervention, that as the reviewers commented “[…] *could lower the threshold for smFRET expertise, allowing for a greater number of scientists to take advantage of this powerful tool”,* rather than prove wrong the existing, robust, software packages developed and operated by experts in the field. We also acknowledge that manual trace selection should not be the standard means to analyse smFRET data, but despite the wide range of available software packages for smFRET data analysis, only some implement advanced automatic sorting of traces. *HAMMY* and *ebFRET* as the reviewers suggested focus on kinetic rate extraction and offer simple thresholds based on intensity and FRET values. Cleaning up traces beyond these simple thresholds often requires extra manual selection. *iSMS* offers more advanced sorting by donor/acceptor intensity, mean FRET and mean stoichiometry for ALEX data as well as automatic detection of photobleaching. *SPARTAN* offers more extensive implementations for automated sorting (26 parameters in total) and is optimised for non-ALEX data. The actual threshold criteria however may vary significantly for each group and experimental system (Fessl et al., 2018; Gouge et al., 2017; Schärfen and Schlierf, 2019; Tsuboyama et al., 2018; Yao et al., 2015). Specialised groups are well trained to navigate through these multiple criteria and accurately define their own, which are optimised to operate on their specific systems (Aznauryan et al., 2016; Fessl et al., 2018; Gouge et al., 2017; Schärfen and Schlierf, 2019; Tsuboyama et al., 2018; Wu et al., 2018; Yao et al., 2015). However the diversity of these sorting criteria might introduce unnecessary bias in an already complicated series of data processing, especially since the advent of commercial instruments has rapidly expanded the smFRET field.

To directly address the comments of the reviewers we performed two types of experimental validations. We firstly compared DeepFRET directly with HAMMY, ebFRET, SPARTAN and iSMS sorting capabilities on simulated data where the ground truth is known. We then compared SPARTAN and iSMS, that have advanced sorting options, on experimental data sets published by other groups.

In the first case, we merged 200 simulated, ground truth smFRET traces with 1800 simulated, nonsmFRET traces (Figure 4—figure supplement 3A and the Materials and methods for FRET distributions and parameter descriptions, respectively) and reverse engineered tif files that would correspond to raw smFRET data. We simulated both ALEX and non-ALEX data as iSMS is optimized for ALEX data while HAMMY, ebFRET and SPARTAN operate optimally on non-ALEX data. The tif files were loaded into the respective software packages and used for extraction and sorting of traces applying a quality score of 0.80 in DeepFRET, default sorting parameters in SPARTAN (except background noise threshold), intensity, stoichiometry and FRET thresholds in iSMS, intensity thresholds in HAMMY and FRET thresholds in ebFRET. We found DeepFRET, SPARTAN and iSMS to recover the underlying ground truth FRET distribution to various levels of detail, while the simple intensity and FRET thresholds of HAMMY and ebFRET would require further sorting for optimal results. Notably, DeepFRET was found to sort traces at least similarly to, or better than, both SPARTAN and iSMS without any parameter tuning (Figure 4—figure supplement 3B). We strongly note that expert users would be able to fine-tune all possible thresholds to better match the ground truth data. However, in a real-world experiment where the ground truth is unknown the task becomes more challenging and finetuning parameters may be subject to bias, especially for non-specialised users. The single threshold-based classification offered by DeepFRET may be crucial for a greater number of scientists to take advantage of this tool.

In the second case, we compared the performance of the three software packages offering advanced sorting on experimental data published by other groups. Selecting non-ALEX and ALEX datasets published by Kilic et al. (Kilic et al., 2018) and Hellenkamp et al. (Hellenkamp et al., 2018), respectively ensures proper testing in diverse FRET settings and data sets. We used practically the default settings in both softwares and cropped data to the first 10 frames of each trace, minimising bleaching without using the default hard threshold at FRET < 0.2 in SPARTAN. Our analysis illustrates that all three software packages are able to reproduce the published FRET distributions from raw tif files with little discrepancy (Figure 4—figure supplement 4). DeepFRET displays equivalent or superior performance to existing sophisticated software packages also on experimental data. We emphasize that existing software packages are very robust and expert users would be able to navigate through and optimize all of the required settings for the individual datasets. The power of DeepFRET is its capacity to analyse both ALEX and non-ALEX data in a reproducible manner only requiring minimal human intervention, and thus minimal expertise in threshold setting, *a*llowing for a greater number of scientists to take advantage of this powerful tool. Acknowledging the lack of comparison to existing software, we have added the new Figure 4—figure supplements 3-4. We have also renamed the sub section “DeepFRET performance on real data,” to “DeepFRET performance on real data, comparison to existing robust softwares for smFRET analysis” and also added a new paragraph in the section explicitly discussing the comparison on simulated ground truth and published data.

3) Along the same lines, a better way to prove DeepFRET's trace analysis power is to take several datasets and compare analyses from HAMMY, ebFRET etc. vs. DeepFRET. The authors should include such comparative analyses on more than one dataset in the revised version of the manuscript.

To address the comment of the reviewers we directly compared the performance of DeepFRET to published results on two experimental datasets (ALEX and non-ALEX) across different groups (Figure 4—figure supplement 4 and answer to comment 2). We found that DeepFRET was able to reproduce the published smFRET distributions using a simple quality threshold of 0.80 without further human intervention, and furthermore so equally or better than existing software, (see also answer to reviewer comment 2). These data further validate the performance and power of DeepFRET as compared to other existing software requiring user-defined thresholds that may require specialised expertise of its users. We have explained the comparison in the main text and added the new Figure 4—figure supplement 4 in the manuscript.

4) It would be helpful if, in the Discussion section, the authors could provide a discussion of the tool's limitations. A discussion that talks about cases where their tool might fail would be useful for researchers who want to use their tool or build upon it. For example, in the discussion, the Materials and methods section notes that photophysical effects that are sometimes observed in smFRET experiments can be problematic for the method (it seems like the tool would likely classify these as non-useful traces even though they might reflect the "true" signal from the experiment [e.g. observation of PIFE in work from TJ Ha's group]).

Following the comment of the reviewers, we added a new paragraph to the Discussion section outlining the limitations of the current version of DeepFRET and what could be done in the future to improve it. As outlined in the previous sections, DeepFRET performs accurately on both simulated and experimental 2-color smFRET data across multiple laboratories. The limitations that are discussed in the revised version of the manuscript can be addressed by expert users through simulation of new training data and re-training of the DNN model following our instructions (https://github.com/hatzakislab/DeepFRET-Model) as described in the Materials and methods section.

References:

Aznauryan, M., Søndergaard, S., Noer, S.L., Schiøtt, B., Birkedal, V., 2016. A direct view of the complex multi-pathway folding of telomeric G-quadruplexes. Nucleic Acids Res. 44, 11024– 11032. doi:10.1093/nar/gkw1010

Fessl, T., Watkins, D., Oatley, P., Allen, W.J., Corey, R.A., Horne, J., Baldwin, S.A., Radford, S.E., Collinson, I., Tuma, R., 2018. Dynamic action of the Sec machinery during initiation, protein translocation and termination. *eLife* 7. doi:10.7554/*eLife*.35112

Gouge, J., Guthertz, N., Kramm, K., Dergai, O., Abascal-Palacios, G., Satia, K., Cousin, P., Hernandez, N., Grohmann, D., Vannini, A., 2017. Molecular mechanisms of Bdp1 in TFIIIB assembly and RNA polymerase III transcription initiation. Nat. Commun. 8, 130.

doi:10.1038/s41467-017-00126-1

Schärfen, L., Schlierf, M., 2019. Real-time monitoring of protein-induced DNA conformational changes using single-molecule FRET. Methods 169, 11–20.

doi:10.1016/j.ymeth.2019.02.011

Tsuboyama, K., Tadakuma, H., Tomari, Y., 2018. Conformational activation of argonaute by distinct yet coordinated actions of the hsp70 and hsp90 chaperone systems. Mol. Cell 70, 722-729.e4. doi:10.1016/j.molcel.2018.04.010

Wu, S., Liu, J., Wang, W., 2018. Dissecting the Conformational Dynamics-Modulated Enzyme Catalysis with Single-Molecule FRET. J. Phys. Chem. B 122, 6179–6187.

doi:10.1021/acs.jpcb.8b02374

Yao, C., Sasaki, H.M., Ueda, T., Tomari, Y., Tadakuma, H., 2015. Single-Molecule Analysis of the Target Cleavage Reaction by the *Drosophila* RNAi Enzyme Complex. Mol. Cell 59, 125–132. doi:10.1016/j.molcel.2015.05.015